# Lipid Raft Destabilization Impairs Mouse TRPA1 Responses to Cold and Bacterial Lipopolysaccharides

**DOI:** 10.3390/ijms21113826

**Published:** 2020-05-28

**Authors:** Justyna B. Startek, Karel Talavera

**Affiliations:** Laboratory of Ion Channel Research, Department of Cellular and Molecular Medicine, KU Leuven; VIB Center for Brain & Disease Research, Herestraat 49, Campus Gasthuisberg O&N1 bus 802, 3000 Leuven, Belgium; justyna.startek@kuleuven.vib.be

**Keywords:** mouse TRPA1, cholesterol, sphingolipids, lipid rafts, LPS, cold

## Abstract

The Transient Receptor Potential ankyrin 1 cation channel (TRPA1) is expressed in nociceptive sensory neurons and epithelial cells, where it plays key roles in the detection of noxious stimuli. Recent reports showed that mouse TRPA1 (mTRPA1) localizes in lipid rafts and that its sensitivity to electrophilic and non-electrophilic agonists is reduced by cholesterol depletion from the plasma membrane. Since effects of manipulating membrane cholesterol levels on other TRP channels are known to vary across different stimuli we here tested whether the disruption of lipid rafts also affects mTRPA1 activation by cold or bacterial lipopolysaccharides (LPS). Cooling to 12 °C, *E. coli* LPS and allyl isothiocyanate (AITC) induced robust Ca^2+^ responses in CHO-K1 cells stably transfected with mTRPA1. The amplitudes of the responses to these stimuli were significantly lower in cells treated with the cholesterol scavenger methyl β-cyclodextrin (MCD) or with the sphingolipids hydrolyzer sphingomyelinase (SMase). This effect was more prominent with higher concentrations of the raft destabilizers. Our data also indicate that reduction of cholesterol does not alter the expression of mTRPA1 in the plasma membrane in the CHO-K1 stable expression system, and that the most salient effect is that on the channel gating. Our findings further indicate that the function of mTRPA1 is regulated by the local lipid environment and suggest that targeting lipid-TRPA1 interactions may be a strategy for the treatment of pain and neurogenic inflammation.

## 1. Introduction

Biological membranes are dynamic and complex lipid–protein systems, forming a protective boundary around the cell and participating in multiple physiological processes [1]. Lipid bilayers, made out of two opposing leaflets, contain multiple, asymmetrically distributed lipids that belong to three classes: glycerophospholipids, sphingolipids and sterols [2,3]. High concentrations of sterols such as cholesterol, induce conformational ordering of the hydrocarbon chains of other lipids, resulting in stiffening and reduced permeability of the membrane [4]. In contrast, cholesterol depletion significantly increases the bilayer permeability and affects the conformation and activities of membrane proteins [5]. Cholesterol influences the membrane thickness and fluidity, as well as the organization of membrane domains, such as lipid rafts [6]. Lipid rafts are small, highly dynamic signaling platforms that support correct localization and function of many proteins. This seems to be particularly important for sensory signaling, as the modification of the cholesterol content at the plasma membrane alters the function of several sensory Transient Receptor Potential (TRP) cation channels [7,8,9,10,11,12]. One of these proteins, the ankyrin-rich TRPA1, is arguably the most versatile polymodal sensor, being activated by thermal [13,14,15,16,17,18,19,20,21,22] and mechanical [15,23,24,25,26,27] stimuli, and by an impressive variety of chemicals [28,29,30].

We recently showed that mouse mTRPA1 localizes mainly in cholesterol rich domains and that disruption of lipid rafts changes the expression pattern of this protein from an extended distribution along the plasma membrane to well-defined clusters [31]. Our data also indicated that mTRPA1 channels could directly interact with cholesterol through cholesterol recognition amino acid consensus (CRAC) motifs located in TM2 and TM4 domains of the channel. Mutation of specific amino acids in these motifs reduces mTRPA1 responses to AITC and decreases the expression of the channel in the plasma membrane [31]. Thus, direct cholesterol-mTRPA1 interactions seem to be necessary for normal channel gating and targeting to the membrane specific subdomains [31]. These findings led us to propose that, together with the other key polymodal nociceptive channel TRPV1 [12], mTRPA1 forms signaling complexes in membrane microdomains, which support direct protein-protein interactions and/or Ca^2+^-dependent crosstalk.

One of the questions generated by our previous study is whether alterations in cholesterol and sphingolipid levels influence the responses of TRPA1 to other types of stimuli. This question does not have an obvious answer, because the influence of cholesterol manipulations on TRP channel function has been shown to be complex. For example, cholesterol depletion was reported not to affect the thermal sensitivity of TRPV1 [7], but later shown to reduce the amplitude of currents activated by capsaicin and protons [8]. In this report, we show that lipid raft disruption reduces the responses of mTRPA1 to cold [14] and bacterial lipopolysaccharides [32,33]. Our findings further support the hypotheses that mTRPA1 function strongly depends on the local lipid environment and suggest that lipid-TRPA1 interactions could be also used as therapeutic targets for the treatment of pain and neurogenic inflammation.

## 2. Results

### 2.1. Cholesterol Depletion Reduces mTRPA1 Response to Cold

We used ratiometric Ca^2+^ imaging to examine the responses of CHO-K1 cells stably expressing mTRPA1 to cold in control or after pretreatment with 1, 5 or 10 mM methyl-β-cyclodextrin (MCD). The responses were analyzed in terms of maximal amplitude of the change in intracellular Ca^2+^ concentration (Δ[Ca^2+^]). We also measured the maximal first time derivative (rising rate) of intracellular Ca^2+^ responses, (d[Ca^2+^]/dt)_MAX_, which is more directly related to the currents carried by the mTRPA1 channels. Control cells responded with an increase in intracellular Ca^2+^ concentration upon application of cooling solution (12 °C; Figure 1a). Neither Δ[Ca^2+^] or (d[Ca^2+^]/dt)_MAX_ of the responses to cold followed a normal distribution, as the majority of cells displayed relatively small and slow responses, resulting in a mismatch between the median and the average values (Figure 1c,e). A subsequent application of 100 µM AITC triggered very robust Ca^2+^ responses (> 500 nM) in the majority of cells (Figure 1a,d), and the corresponding distribution of amplitudes was approximated by a normal distribution (see the match between the mean and median values in Figure 1d). 

Cells pretreated with 1 mM MCD responded to cooling with amplitudes similar to those of control cells, but cells pretreated with 5 mM or 10 mM MCD responded with significantly smaller amplitudes (Figure 1b,c). A reduction of the maximal values of the first derivative was already apparent from 1 mM MCD and was stronger at 5 and 10 mM (Figure 1e). As expected from our previous study [31], the MCD treatments also decreased the amplitude and maximal rising rate of responses to the TRPA1 electrophilic agonist AITC [34,35] (Figure 1b,d,f).

Next, we aimed at evaluating whether cholesterol depletion alters the response of mTRPA1 channels to cold, independently of an eventual effect on channel expression at the plasma membrane [31]. A way to do this could be normalizing the response of each cell to cold by its response to AITC. If the normalized amplitudes are changed by increasing MCD concentrations, this would mean that there is a direct effect of cholesterol depletion on channel gating. However, this method would be appropriate only if the amplitudes of the responses to cold are strictly correlated to the amplitudes of the responses to AITC across the cell population for control and all MCD pretreatment conditions. We found that in control cells the amplitudes of the responses to cold and AITC showed some degree of correlation (Pearson’s correlation coefficient, *R* = 0.72). However, the correlation was low in cells treated with MCD (*R* = 0.41, 0.54 and 0.20, for MCD 1, 5 and 10 mM, respectively). These observations are consistent with the weak action of these stimuli in cells treated with MCD [36]. Another way to contrast the effects of cholesterol depletion on cold- and AITC-induced responses is to compare the levels of statistical significance of the effects of the MCD treatments (Figure 1c–f). These indicate that the sensitivities of the responses to cold and AITC to the MCD treatments were rather similar.

To further investigate mTRPA1 activation after disruption of lipid rafts, we treated the cells with sphingomyelinase (SMase). This enzyme hydrolyzes sphingomyelin (SM), which is one of the components of lipid rafts and roughly constitutes 2%–15% of total lipids of cellular membranes [37,38]. SM hydrolysis triggers the formation of ceramide and phosphocholine, inducing an increase in the membrane tension [38,39]. An intermediate response to the bilayer ordering is release of cholesterol and further reorganization of the lipid rafts [40]. Treatment of cells with SMase reduced the amplitude as well as the maximal rising rate of the response to cold in a concentration dependent manner (12 °C; Figure 2a–c,e). Additionally, the amplitude and the maximal rising rate of responses to AITC application were reduced by SMase (Figure 2b,d,f), as predicted from our previous report [31].

Like for the MCD treatment, the levels of statistically significant differences indicate that the sensitivities of the responses to cold and AITC to the SMase treatments were rather similar.

### 2.2. Cholesterol Depletion Reduces mTRPA1 Response to Bacterial Lipopolysaccharides

Extracellular application of *Escherichia coli* (*E. coli*) LPS (20 µg/mL) triggered intracellular Ca^2+^ responses (Figure 3a,c), as previously reported [32,33]. The amplitudes of LPS-induced responses did not follow a normal distribution, but were distributed in two groups, one with small values and the other with amplitudes larger than 0.75 µM (Figure 3c). The maximal rising rate showed a similar trend (Figure 3e). Pretreatment with MCD 1 or 5 mM did not result in statistically significant changes in these magnitudes, but a modest reduction was found in cells pretreated with 10 mM MCD (Figure 3b,c,e). The responses to AITC were confirmed to be reduced by 1, 5 and 10 MCD (Figure 3b,d,f).

Next, we determined the effects of pretreatments with SMase (1–50 mUN) on the responses to LPS. We found statistically significant decrease in the amplitude and the maximal rising rate of the responses to LPS only for the highest concentration (Figure 4a–c,e). This contrasts with the significant effects on the responses to AITC, which were found from 10 mUN for the amplitudes (Figure 4d) and from 1 mUN for the maximal rising rate (Figure 4f). Taken together our results show that the MCD and SMase treatments had a smaller influence on the action of LPS than on those of cold and AITC. 

### 2.3. Cholesterol Depletion Reduces Sensitivity but Not the Maximal Response of mTRPA1 to AITC

The weaker effects of lipid raft disruption on the responses to LPS cannot be explained by a decrease in channel expression as a main effect of cholesterol depletion because this would affect the action of all stimuli in similar extents. In turn, this raised the question of whether such an effect actually occurs in CHO-K1 cells, as we have previously documented in HEK293T cells [31]. To test this, we performed another series of experiments, to compare the concentration-dependence curves for AITC determined in the control and after pretreatment with 10 mM MCD in mTRPA1 CHO-K1 cells, and to see whether the treatment reduces the maximal response to this compound. We tested AITC at concentrations up to 200 µM because at higher values we have seen an inhibitory effect [41]. In control cells AITC triggered Ca^2+^ responses in a concentration-dependent manner, with an EC_50_ of 4.6 ± 1.1 µM (Figure 5a-left,b). In cells pretreated with MCD the EC_50_ was 15-fold higher (Figure 5a-right,b), thus qualitatively similar to the effect reported in our previous study in HEK293 cells [31]. However, the estimated maximal responses to AITC were not different between control and MCD-treated cells (1.32 ± 0.09 µM and 1.4 ± 0.3 µM, respectively). This indicates that the expression of mTRPA1 is not altered by the reduction of cholesterol in the CHO-K1 stable expression system, and that the most salient effect is that on the channel gating.

## 3. Discussion

The survival of living organisms vastly relies on a constant monitoring of the external and inner chemical environments, allowing for adequate protective reactions against harmful stimuli. This is particularly important for the detection of noxious chemicals, changes in temperature and osmolarity and mechanical stimuli. TRPA1 plays an important role in the detection of harmful thermal, mechanical and chemical stimuli [28,29]. For instance, TRPA1 is activated by electrophilic compounds such as AITC and cinnamaldehyde, through a mechanism that involves covalent modification of cysteine residues [34,35]. This channel is also activated by a large group of non-electrophilic chemicals that may induce mechanical perturbations upon insertion in the plasma membrane [28]. Recently, TRPA1 activation was associated with channel’s ability to sense changes in the order of the surrounding bilayer [33]. It has been proposed that bacterial LPS activated TRPA1 [32,33,42] in non-canonical way, by inducing changes in the mechanical properties of the membrane, which may be detected by the channel [33]. Multiple TRPA1 agonists such as trinitrophenol and chlorpromazine [43], primary alcohols [44], parabens [45], phenolic compounds [46], farnesyl thiosalicylic acid and its analogs [47], 6-gingerol analogs [48,49], menthol [50] and camphor [51] are also able to partition into the plasma membrane, thereby inducing alterations in the channel local environment, such as changes in the curvature, tension and/or thickness.

Regardless of the growing indications that membrane properties and composition are crucial for TRPA1 activation by multiple chemicals, the mechanism of channel activation remains elusive. TRPA1 function and localization pattern has been linked to cholesterol levels in the membrane. This channel was shown to be expressed in specific membrane domains characterized by high concentrations of specific lipids such as cholesterol, gangliosides and sphingolipids [11,31,52]. Modification of the composition of lipid rafts by bilayer destabilizers, such as MCD and SMase, strongly reduces mTRPA1 chemosensitivity to electrophilic (AITC) and non-electrophilic (thymol) agonists [31]. However, it remained unknown whether modifications of cholesterol and sphingolipids levels also affect the response of mTRPA1 to stimuli of distinct nature. In this study we present data showing that this is indeed the case, although to different extents, for the activation of by cold and LPS.

The fact that alterations of the membrane composition affect mTRPA1 responses to very distinct stimuli (chemical: AITC and thymol [31] and physical: cold and LPS, present results) further supports the idea that artificial expression systems may be of limited reach in the understanding of channel structure and function [31]. In a similar line, variations in the lipid environment of TRPA1 across expression systems or in the presence of lipid-channel interaction sites might contribute to the distinct functional properties of isoforms reported in the literature. In this sense, as done previously in structure-function analyses (see for review [13,28,29,53]), future studies may test the role of lipid-channel interactions exploiting the fact that, depending on the species, TRPA1 channels can be activated by heat [13,16,54,55,56,57,58], by cold [14,17,18,57] or insensitive to cold [13,59]. Of particular interest is the new technical possibility of studying TRPA1 in artificial membranes [16,17,25], which in principle may allow determining the influence of individual lipid components on the membrane expression pattern and the functional properties of the channel.

Regarding the mechanism underlying the reduction of mTRPA1 responses by cholesterol depletion, it was previously shown that treatment with MCD induced a dual action in HEK293T cells: a decrease in the sensitivity for stimulation with AITC (increased EC_50_) and a reduction in the number of channels expressed at the plasma membrane. A major finding of the present study is that the latter effect does not seem to occur in the CHO-K1 expression system, as we found that the responses to AITC at high concentrations we nearly identical. The reason for this difference remains unclear, but could be related to the type of heterologous expression, transient vs. stable transfection, as in the former the levels of mTRPA1 expression are higher and could therefore be more liable to a decrease by the change in membrane composition.

The present experiments did, however, recapitulate the decrease in sensitivity to AITC, i.e., a 15-fold increase in the EC_50_, which is actually larger than the 5-fold increase found previously in HEK293T cells [31]. Thus, we conclude, first, that the most consistent effect of cholesterol depletion on mTRPA1 was the reduction of the channel’s sensitivity to stimulation, and second, that the reduction in the responses to cold and LPS we reported here cannot be explained by a decrease in the channel expression at the plasma membrane. 

Interestingly, cholesterol depletion was reported to produce an opposite effect on TRPM8, enhancing the responses of this channel to cold [9]. The mechanisms underlying the regulation of TRPM8 by cholesterol remain unknown, but were discussed in terms of possible influence of changes in membrane fluidity, direct interactions with raft-specific lipids and channel regulation by other membrane proteins confined to cholesterol-enriched membrane domains [9]. It could be argued that the fluidizing effect that MCD and SMase have on the membrane may counteract the rigidifying effect of cold. This might reduce the ability of cold to stimulate mTRPA1, as the increase in membrane order was recently related to the activation of this channel [23,33].

As for the stimulation of mTRPA1 by LPS, the disruption of lipid rafts may impair the insertion of LPS in the membrane. Indeed, it has been shown that cholesterol plays a key role in LPS-membrane interactions, and that LPS has high binding affinity to phosphatidylcholine-, sphingomyelin- and cholesterol-containing artificial membranes that mimic lipid rafts [60,61]. Another salient result of our present study is that the stimulation of mTRPA1 by LPS was more resistant to the treatments with MCD and SMase, as the effects were observed at higher concentration than for cold and AITC. This finding further supports the idea that cholesterol depletion does not significantly affect mTRPA1 expression in CHO-K1 cells, but reduces the sensitivity of the channels to the different stimuli in distinct manners.

Interestingly, the role of lipid rafts in the interactions that mediate the invasion of host cells by pathogenic microorganisms have been extensively studied over the last few years [62], and it is known that the canonical LPS detection machinery in immune cells is supported by raft clustering of signaling proteins such as Toll-like receptor 4 (TLR4) [63,64], GPI-anchored proteins [65], for example CD14 [66] and other pro-inflammatory protein complexes [63]. The similarities of the membrane contexts required by both the LPS-induced TRPA1 activation and the well-established TLR4-mediated signaling constitutes further support to the idea that TRPA1 indeed functions as a detector of endotoxins, triggering fast defense mechanisms against gram-negative bacterial infections mediated by sensory neurons [32].

Taken together, our findings support the hypotheses that mTRPA1 activation by multiple stimuli, including electrophilic and non-electrophilic agonists, as well as cold and bacterial endotoxins, depends on the local lipid environment. The role of sensory TRP channels in the induction of pain and neurogenic inflammation in the context of bacterial infections [23,27,32,67] suggest them as possible direct therapeutic targets. However, our data, together with previous findings indicating that cholesterol-lowering drugs improve neuropathic pain [68], suggest that lipid-TRPA1 channel interactions could be also used in the treatment of infections.

## 4. Materials and Methods

### 4.1. Cell Culture

Chinese hamster ovary (CHO-K1) cells from the American Type Culture Collection were grown in DMEM containing 10% fetal bovine serum, 2% glutamax (Gibco/Invitrogen), 1% non-essential amino acids (Invitrogen) and 200 µg/mL penicillin/streptomycin at 37 °C in a humidity-controlled incubator with 5% CO_2_. As mTRPA1 expression system we used CHO-K1 cells stably transfected with mTRPA1 [18].

### 4.2. Disruption of Lipid Rafts with MCD or SMase

Cells were washed with serum-free culture medium and incubated with different concentrations (1–10 mM) of methyl-β-cyclodextrine (MCD; Sigma-Aldrich) or sphingomyelinase (SMase; 1–50 mUN) from *Bacillus cereus* (Sigma-Aldrich) for 1 h at 37 °C in a humidity-controlled incubator. According to previous studies [69,70,71], none of these treatments are expected to induce significant cytotoxic effects, which we confirmed by inspecting the cell morphology.

After these treatments the cells were washed and used for intracellular Ca^2+^ imaging experiments within 5 min. None of our experimental solutions contained serum, which would otherwise support de-novo cholesterol synthesis. Although the rate of cholesterol redistribution in native membrane bilayers remains unknown, we are confident that we do induce reduction of cholesterol and sphingomyelin levels, as many other studies reports [72,73,74,75,76].

### 4.3. Ratiometric Intracellular Ca^2+^ Imaging

For intracellular Ca^2+^ imaging experiments cells were incubated with 2 µM Fura-2 AM (Biotium, Hayward, CA, USA) for 40 min at 37 °C in a humidity-controlled incubator. Fluorescence was measured with alternating excitation at 340 and 380 nm using a monochromator-based imaging system consisting of an MT-10 illumination system (Tokyo, Japan) and Cell^M^ software from Olympus. All experiments were performed using the standard Krebs solution (see above) at 25 °C. Fluorescence intensities were corrected for background signal and presented as the ratio F340/F380 from which intracellular Ca^2+^ concentration was calculated as described previously [77]. Data were analyzed and presented as mean ± s.e.m. using Origin 9.0 (OriginLab Corporation).

### 4.4. Data and Statistical Analysis

If not stated otherwise, the non-parametric Kolmogorov–Smirnov test was used to assess statistical significance. Most results are shown using notched box plots in which the data is presented as individual values (scatter data symbols), along with the median (position of the notch center), the mean (horizontal line), the 25 and 75 percentiles (vertical box limits) and the 95% confidence interval of the median (notch). Asterisks represent the significance (* *p* < 0.05; ** *p* < 0.01; *** *p* < 0.001) and *n* denotes the sample size. The concentration-dependence of AITC effects is shown as mean ± s.e.m.

The concentration-dependence curves for the stimulatory effect of AITC on mTRPA1 were fit by a Hill function of the form:$$\Delta \left[Ca^{2+}\right]=\frac{\mathrm{Max}\;\times {\left[AITC\right]}^{H}}{{\left[AITC\right]}^{H}+EC_{50}^{H}}$$
where Max is the maximal increase in intracellular Ca^2+^ levels obtained at high concentrations of AITC ([AITC]); *EC_50_* is the half effective concentration and *H* is the Hill coefficient.

## Figures and Tables

**Figure 1 ijms-21-03826-f001:**
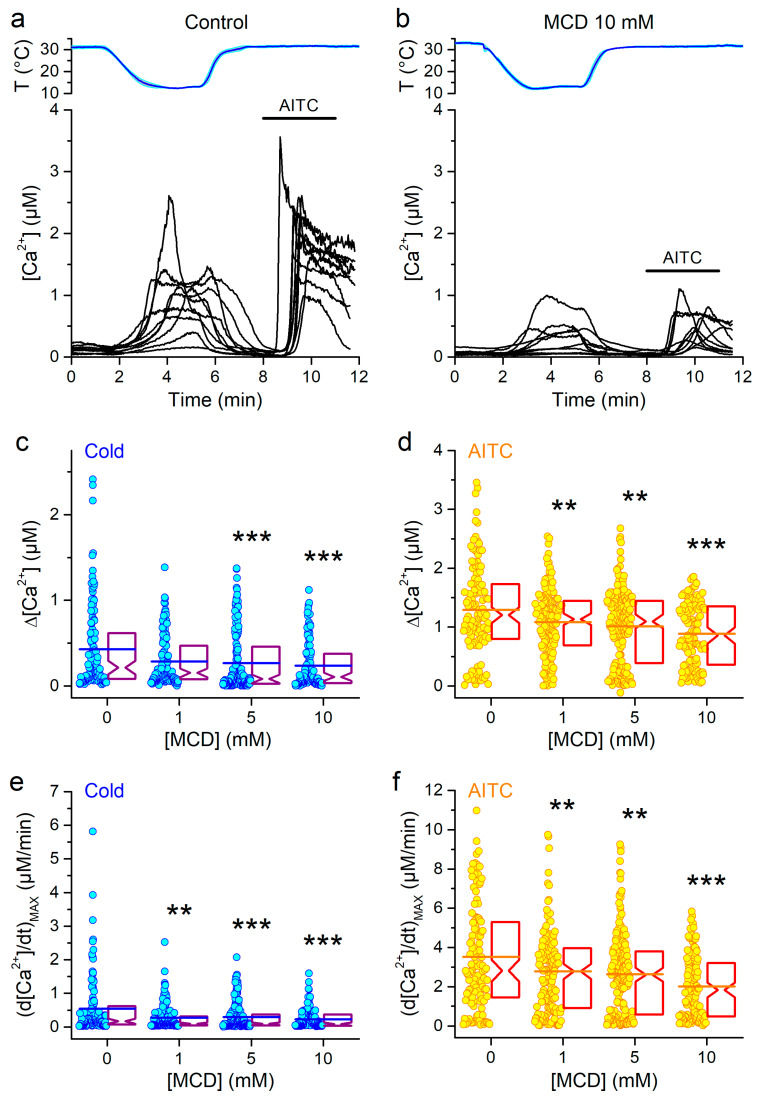
Effects of cholesterol depletion on cold-induced responses of mTRPA1. (**a** and **b**) Representative traces of [Ca^2+^] change induced by cooling (upper panels) in the control condition (**a**) and after pretreatment with 10 mM MCD (**b**). Application of AITC (100 µM) was used as control for mTRPA1 activation. (**c** and **d**) Amplitudes of [Ca^2+^] responses evoked by cold (**c**) and AITC (**d**) in control condition (*n* = 118) or after pretreatment with 1 (*n* = 149), 5 (*n* = 202) or 10 mM (*n* =130) of MCD. (**e** and **f**) Maximal amplitudes of the first time derivative of the intracellular Ca^2+^ signal elicited by cold (**e**) and AITC (**f**) in control condition or after treatment with 1, 5 or 10 mM of MCD (the *n* numbers are the same as in panels c and d). The symbols *, ** and *** indicate *p* < 0.05, *p* < 0.01 and *p* < 0.001, respectively; Kolmogorov–Smirnov test.

**Figure 2 ijms-21-03826-f002:**
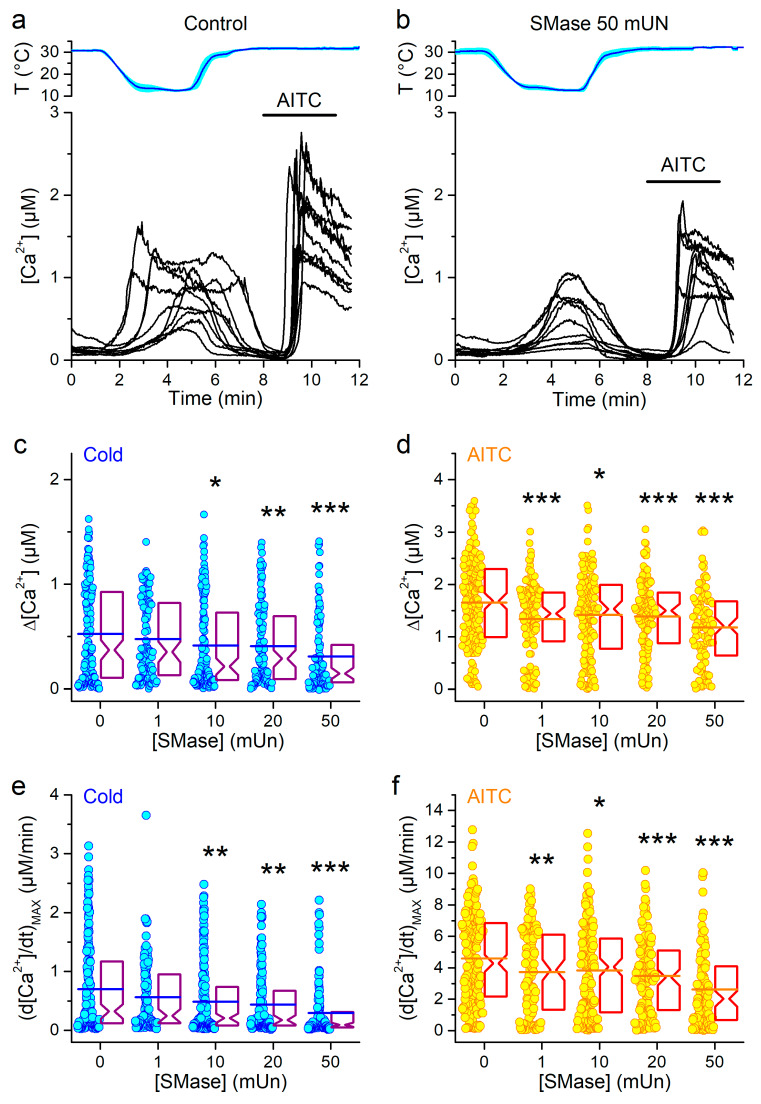
Effects of SMase pretreatment on mTRPA1 responses to cold. (**a** and **b**) Representative traces of [Ca^2+^] change induced by cold in the control condition (**a**) and after pretreatment with 50 mUN SMase (**b**). Application of AITC (100 µM) was used as control for mTRPA1 activation. (**c** and **d**) Amplitudes of [Ca^2+^] responses evoked by cold (**c**) and AITC (**d**) in control condition (*n* = 183) or after pretreatment with 1 (*n* = 126), 10 (*n* = 164), 20 (*n* = 135) or 50 mUN (*n* = 111) of SMase. (**e** and **f**) Maximal amplitudes of the first time derivative of the intracellular Ca^2+^ signal elicited by cold (**e**) and AITC (**f**) in control condition or after treatment with 1, 10, 20 or 50 mUN of MCD (the *n* numbers are the same as in panels c and d). The symbols *, ** and *** indicate *p* < 0.05, *p* < 0.01 and *p* < 0.001, respectively; Kolmogorov–Smirnov test.

**Figure 3 ijms-21-03826-f003:**
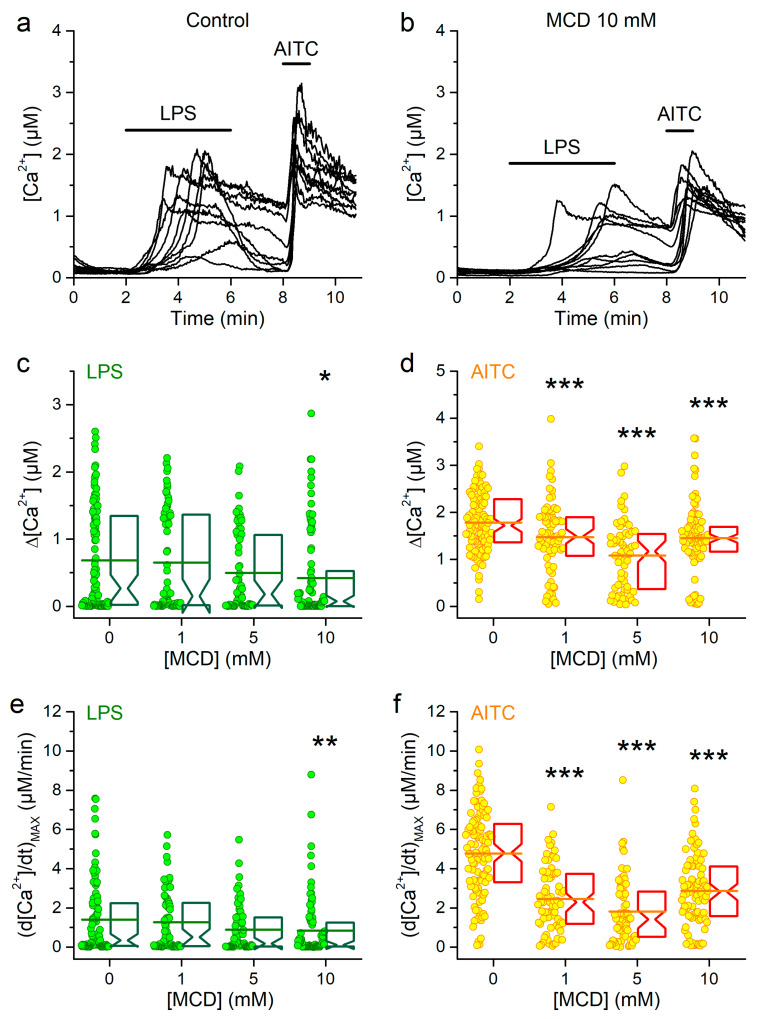
Effects of cholesterol depletion using MCD on mTRPA1 responses to LPS. (**a** and **b**) Representative [Ca^2+^] traces showing the effects of *E. coli* LPS (20 µg/mL) in control condition (**a**) and after pretreatment with 10 mM MCD (**b**). Application of AITC (100 µM) was used as control for mTRPA1 activation. (**c** and **d**) Amplitudes of [Ca^2+^] responses to LPS (**c**) and AITC (**d**) in control (*n* = 102) or after pretreatment with 1 (*n* = 72), 5 (*n* = 64) or 10 mM (*n* = 90) of MCD. Maximal amplitude of the first time derivative of the intracellular Ca^2+^ signal elicited by LPS (**e**) and AITC (**f**) in control condition or after treatment with 1, 5 or 10 mM of MCD (the *n* numbers are the same as in panels c and d). The symbols *, ** and *** indicate *p* < 0.05, *p* < 0.01 and *p* < 0.001, respectively; Kolmogorov–Smirnov test.

**Figure 4 ijms-21-03826-f004:**
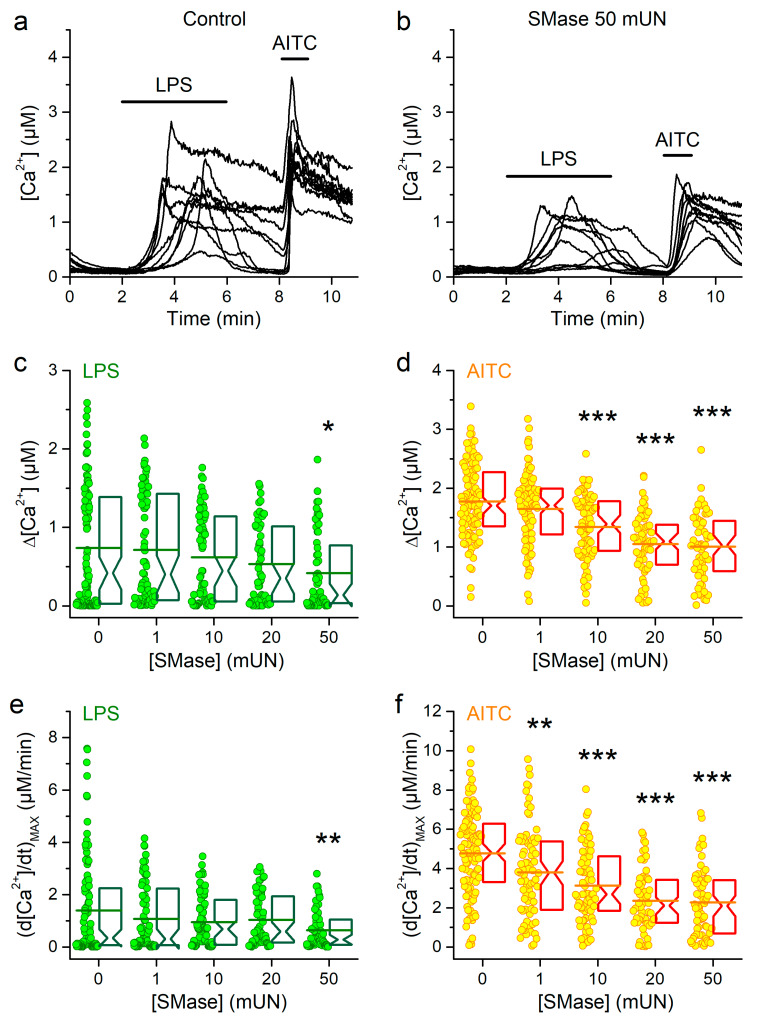
Effects of SMase pretreatment on mTRPA1 responses to LPS. (**a** and **b**) Representative [Ca^2+^] traces showing the effects of *E. coli* LPS (20 µg/mL) in control condition (**a**) and after pretreatment with 50 mUN SMase (**b**). Application of AITC (100 µM) was used as control for mTRPA1 activation. (**c** and **d**) Average amplitude of [Ca^2+^] responses to LPS (**c**) and AITC (**d**) in control condition (*n* = 108) or after pretreatment with 1 (*n* = 83), 10 (*n* = 75), 20 (*n* = 60) or 50 mUN (*n* = 64) of SMase. Maximal amplitude of the first time derivative of the intracellular Ca^2+^ signal elicited by LPS (**e**) and AITC (**f**) in control condition or after treatment with 1, 10, 20 or 50 mUN of SMase (the *n* numbers are the same as in panels c and d). The symbols *, ** and *** indicate *p* < 0.05, *p* < 0.01 and *p* < 0.001, respectively; Kolmogorov–Smirnov test.

**Figure 5 ijms-21-03826-f005:**
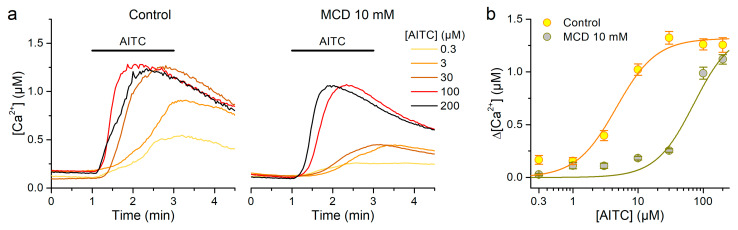
Effects of cholesterol depletion using MCD on mTRPA1 activation by AITC in stably transfected CHO-K1 cells. (**a**) Mean traces of Ca^2+^ responses to different concentrations of AITC in control (left panel) or after treatment with 10 mM MCD (right panel; *n* = 63–178). (**b**) Dose-response curves of the amplitude of the Ca^2+^ responses to AITC averaged over different cells. The lines represent fits with Hill functions with parameters EC_50_ = 4.6 ± 1.1 µM and 70 ± 20 µM, Δ[Ca^2+^]_MAX_ = 1.32 ± 0.09 µM and 1.4 ± 0.3 µM and H = 1.4 ± 0.3 and 1.5 ± 0.5, for control and MCD 10 mM, respectively.

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
