# Peer review of "Lipid Raft Destabilization Impairs Mouse TRPA1 Responses to Cold and Bacterial Lipopolysaccharides"

_ijms, 2020, doi:10.3390/ijms21113826_

Round 1

Reviewer 1 Report

In their MS, Startek and Talavera provide evidences that lipid raft destabilization via cholesterol and sphingomyelin depletion results in decreased TRPA1 activity induced by cold temperature and bacterial lipopolysaccharide (LPS).

The results presented in this Communication are important extensions of their earlier publication in which authors proved the role of lipid rafts in TRPA1 activation by the canonical chemical agonist AITC. Since certain ligands can activate TRP channels with specific mechanisms differing from the action of other activators and modality specific inhibitors are one of the leading trends in the current development of drugs targeting nociceptive thermosensitive TRP channels, e.g. TRPV1, it is an important goal to describe ligand/activator specificity in molecular regulation in TRP channel activation. TRPA1 is an especially important target as a mediator of various noxious stimuli involving cold, various chemicals and membrane perturbances, as detailed in the paper. Therefore authors investigated whether lipid raft disruption (either by cholesterol or sphingomyelin depletion) affects TRPA1 activation by cold and LPS, as well. The results clearly indicate, that the activating effect of cold and LPS, similar to AITC, are significantly decreased following cholesterol (by methyl β-cyclodextrin - MCD) or sphingomyelin (by sphingomyelinase - SMase) depletion. The results are convincing and important, although I have a few comments and suggestions to discuss which could further support the conclusions:

MCD and SMase pretreatment was applied before the experiments. What was the time interval between the pretreatments and the experiments? Is (a partial) “re-ordering” of lipid rafts possible in this time? Can these pretreatments influence cellular viability which might affect the results? Would not be possible to test the effect of cholesterol/sphingomyelin depletion on the same cell, i.e. challenge the same cell before and after MCD/SMase application? The decrease of the amplitudes of the Ca2+ signals is clearly evidenced. Did MCD and SMase pretreatments also influence the slope of the signals? Instead of n>50, a more exact range or sample size should be given.

Author Response

In their MS, Startek and Talavera provide evidences that lipid raft destabilization via cholesterol and sphingomyelin depletion results in decreased TRPA1 activity induced by cold temperature and bacterial lipopolysaccharide (LPS).

The results presented in this Communication are important extensions of their earlier publication in which authors proved the role of lipid rafts in TRPA1 activation by the canonical chemical agonist AITC. Since certain ligands can activate TRP channels with specific mechanisms differing from the action of other activators and modality specific inhibitors are one of the leading trends in the current development of drugs targeting nociceptive thermosensitive TRP channels, e.g. TRPV1, it is an important goal to describe ligand/activator specificity in molecular regulation in TRP channel activation. TRPA1 is an especially important target as a mediator of various noxious stimuli involving cold, various chemicals and membrane perturbances, as detailed in the paper. Therefore authors investigated whether lipid raft disruption (either by cholesterol or sphingomyelin depletion) affects TRPA1 activation by cold and LPS, as well. The results clearly indicate, that the activating effect of cold and LPS, similar to AITC, are significantly decreased following cholesterol (by methyl β-cyclodextrin - MCD) or sphingomyelin (by sphingomyelinase - SMase) depletion. The results are convincing and important, although I have a few comments and suggestions to discuss which could further support the conclusions:

Ans/ We very much appreciate the positive comments of this Reviewer on the value of our study.

Questions

MCD and SMase pretreatment was applied before the experiments. What was the time interval between the pretreatments and the experiments?

Ans/ Cells were treated for 1 h and used for experiments within 5 min after removal of the MCD- or SMase-containing solutions. This is now indicated in lines 286-287.

Is (a partial) “re-ordering” of lipid rafts possible in this time?

Ans/ We had two reasons not to keep MCD and SMase during the running of the experiments. First, the presence of these agents may have a direct impact on channel function. Second, the constant perfusion we apply to the cells would make the experiments very costly.

Nevertheless, in our experiments all treatments and solutions contained no serum, which would otherwise support de-novo cholesterol synthesis. Thus, lipid rafts components could be only restored from remaining internal stores or re-ordering of remaining in the membrane raft constituents (around 40% of cholesterol would remain intact after 1 h treatment with 10 mM MCD treatment, according to previous studies (Mahammad and Parmry, 2014, Ohtani et al., 1989, Kilsdonk et al., 1995, Beseničar et al., 2007). It has been reported that cholesterol diffuses across artificial lipid bilayers in time scales of seconds or less (Lange et al., 1981, Muller and Herrmann, 2002, Gu et al., 2019), minutes (Rodriguez et al., 1995, Schroeder et al., 1996, Haynes et al., 2000, Leventis and Silvius, 2001), or hours (Brasaemle et al., 1988, Rodrigueza et al., 1995). However, the study of the cholesterol flip-flop mechanism in the natural membranes is very difficult as they are highly heterogeneous and dynamic. Thus, the rate of cholesterol redistribution in native membrane bilayers remains unknown. Despite this caveat, we are confident that we do induce reduction of cholesterol and sphingomyelin levels, as many other studies reports (Beseničar et al., 2007, Mahammad and Parmry, 2014, Kilsdonk et al., 1995, Szőke et al., 2010). We include now a short version of this explanation in the Materials and Methods section 4.2. Disruption of lipid rafts with MCD or SMase (lines 287-290).

Can these pretreatments influence cellular viability which might affect the results?

Ans/ Both MCD and SMase treatments can indeed induce cytotoxic effects, such as apoptotic cell death, mitochondrial dysfunction, and changes in cell morphology and proliferation. However, these effects are visible after extended treatments (longer than 1 h) and at concentrations higher than 10 mM for MCD and above 50 mUN for SMase (Kilsdonk et al., 1995; Yancey et al., 1996, Christian et al., 1997). To avoid cell injury during experiments and to select the most appropriate MCD or SMase concentration and time of the treatment we inspected the morphology of cells after the treatments. We decided to use 10 mM of MCD and 50 mUN SMase treatment for 1 h in cell lines as these treatments did not induced toxicity and were described to remove up to 60% of membrane cholesterol. We now make a brief statement on this in the Materials and Methods section 4.2. Disruption of lipid rafts with MCD or SMase (lines 284-285).

Would not be possible to test the effect of cholesterol/sphingomyelin depletion on the same cell, i.e. challenge the same cell before and after MCD/SMase application?

Ans/ We presume that this Reviewer proposes this experiment with the aim to compare the effects of both treatments on the TRPA1-mediated responses in the same cells. There are two elements that reduce our drive to perform these experiments. The first one is that these consecutive treatments will very likely have an impact on cell viability. The second is of technical character. The cells would be on the setup for at least 1 h, without control of CO2 levels, which may also have an impact on cellular viability.

However, the main objection to this experiment would be that the actions of MCD and SMase are not completely independent, because SMase has as secondary effect the reduction of cholesterol upon the rearrangement of membrane lipids that follows the removal of sphingomyelin. Thus, the results of these experiments would not have a straightforward interpretation.

The decrease of the amplitudes of the Ca2+ signals is clearly evidenced. Did MCD and SMase pretreatments also influence the slope of the signals?

Ans/ As suggested we now report the maximal first time derivative during stimulation, (d[Ca2+]/dt)MAX, in addition to the response amplitudes. We explain this in lines 67-70, and commented on those data all along the manuscript were appropriate (indicated in red text). The results found for (d[Ca2+]/dt)MAX are in complete accordance with those already shown with Δ[Ca2+].

Instead of n>50, a more exact range or sample size should be given.

Ans/ As requested, the numbers of cells have been added in the figure legends.

Reviewer 2 Report

The manuscript is well written and contains some interesting information. But these authors recently published a  paper in E-life about lipids and TRPA1 interactions (https://www.ncbi.nlm.nih.gov/pmc/articles/PMC6590989/). There they showed that TRPA1 was located preferably in cholesterol-rich domains and identify cholesterol recognition amino acid consensus. Finally, they concluded that the interaction of TRPA1 with cholesterol was necessary for normal channel gating. In this manuscript, they confirmed these data again. The difference with previously published data only in stimuli applied to TRPA1. The question is what is the novelty of this manuscript? Test system and the conclusion are the same (even the Figures are very similar).  Therefore in present form manuscript does not contain significant additional information to that was published by authors recently.  Cholesterol plays a significant role in channel function ( as they showed before). When they change membrane cholesterol levels, it could affect response to any activation stimuli. It is especially evident for the stimuli that change of membrane plasticity (cold, LPS).

Author Response

The manuscript is well written and contains some interesting information. But these authors recently published a paper in E-life about lipids and TRPA1 interactions (https://www.ncbi.nlm.nih.gov/pmc/articles/PMC6590989/). There they showed that TRPA1 was located preferably in cholesterol-rich domains and identify cholesterol recognition amino acid consensus. Finally, they concluded that the interaction of TRPA1 with cholesterol was necessary for normal channel gating. In this manuscript, they confirmed these data again. The difference with previously published data only in stimuli applied to TRPA1. The question is what is the novelty of this manuscript? Test system and the conclusion are the same (even the Figures are very similar).  Therefore in present form manuscript does not contain significant additional information to that was published by authors recently.  Cholesterol plays a significant role in channel function ( as they showed before). When they change membrane cholesterol levels, it could affect response to any activation stimuli. It is especially evident for the stimuli that change of membrane plasticity (cold, LPS).

Ans/ We provide below several elements that answer this Reviewer’s question of whether our study has sufficient novelty to be published in the IJMS.

First, as noted by Reviewer 1, and as we already explain in our manuscript, the TRP channel literature (e.g., TRPV1) gives many examples of effects of cholesterol depletion that are stimulus-dependent.

Second, as we explain in the Discussion, the effects of LPS and cold on membrane mechanical properties are similar to those of cholesterol (they all increase local membrane rigidity). Hence, the result of the reduction of cholesterol and the application of LPS or cold cannot to be predicted.

Third, it is currently unknown whether the electrophile AITC, cold and LPS induce similar TRPA1 structural changes leading to pore opening. Thus, it is impossible to know a priori if the binding of cholesterol to the channel has the same influence on activation by these three very distinct stimuli. These elements argue in favor of the fairness of the question of whether decreasing cholesterol concentrations alter the responses to LPS or cold, which ought to be answered experimentally.

Fourth, please note that the cellular system we employed in the present study for heterologous expression of TRPA1 (stable expression in CHO-K1 cells) is not the same as the one we used in our eLife paper (transient transfection in HEK293 cells). The former is probably the most popularly used amongst heterologous expressions systems to study TRPA1. Furthermore, following a suggestion of Reviewer #3 we evaluated the effects of cholesterol depletion on the responses to AITC in CHO-K1 cells and found that this induced an increase in the EC50 (as reported in our previous study in HEK293 cells), but not a change in the maximal response to AITC. This indicates that the expression of TRPA1 is not altered by the reduction of cholesterol in the CHO-K1 stable expression system, and that the most salient effect is that on the channel gating. These data introduce an extra element of novelty, and definitely justifies testing the effects of cholesterol reduction on the action of channel stimuli of distinct nature.

Finally, this Reviewer also comments on the fact that the figures are very similar to those of our eLife paper. We do not know if this has to be taken as a criticism, but nevertheless, to address this comment we changed the graphs into a style that is more informative. We now show data + notched box plots in which the reader can appreciate several aspects of the data, including all data points, the mean, the median, the 25 and 75 percentiles and the 95% confidence interval of the median. This is now explained in the Materials and Methods in section Data and statistical analysis. We hope this Reviewer finds the new style and description of the data presentation more satisfactory than the previous one.

Reviewer 3 Report

This rather simple study aims to determine if depletion of membrane cholesterol levels can alter TRPA1 mediated calcium transients in response to cold or LPS stimulation.  It is a follow-up to a study that demonstrated TRPA1 localizes to lipid rafts and thus may be modulated by alterations in the lipid composition of the plasma membrane.  The manuscript is well-written and straight forward.  The manuscript’s main weakness, however, relates to the significance of the findings in this paper, given the results presented in the earlier manuscript.

In the previous paper, the authors demonstrated that cholesterol depletion decreases the expression of TRPA1 on the plasma membrane. Thus, the results of the present study are not a surprise; if the expression of the receptor is decreased the response to any agonist would be decreased.  This would also apply to the response to “physical stimuli”, as the authors classify cold and LPS.  Therefore, I’m unsure how the data reported in this manuscript is novel or significant.  The authors attempt to suggest that other mechanisms may play a role, such as binding of LPS to the plasma membrane and association with lipid rats, which would increase its proximity to TRPA1.  Again, this does not matter if the expression of the channel is also decreased.  The question that I, and most readers, would be interested in is if cholesterol depletion alters the single channel response in response to cold or LPS. This is probably best demonstrated using patch clamp.  However, this might also be accomplished by normalizing each cell's response to cold or LPS by its response to AITC, which would correct for the decrease in receptor expression.  If the normalized signal in response to cold or LPS still decreases, we can hypothesize that there is a direct effect of cholesterol depletion on the channel itself.  In all of the figures, the amplitude of the effects do not seem to match up with the reported change in intracellular calcium in the bar graphs. For example, in Figure 1, the amplitude of the calcium transients appears to all be above 1uM, yet the average reported change in calcium around 0.4uM.  The authors should show traces that better represent the reported averages.  The authors should list the exact number of cells for each experiment instead of the general “n>50” listing they have now.

Author Response

This rather simple study aims to determine if depletion of membrane cholesterol levels can alter TRPA1 mediated calcium transients in response to cold or LPS stimulation.  It is a follow-up to a study that demonstrated TRPA1 localizes to lipid rafts and thus may be modulated by alterations in the lipid composition of the plasma membrane.  The manuscript is well-written and straight forward.  The manuscript’s main weakness, however, relates to the significance of the findings in this paper, given the results presented in the earlier manuscript. In the previous paper, the authors demonstrated that cholesterol depletion decreases the expression of TRPA1 on the plasma membrane. Thus, the results of the present study are not a surprise; if the expression of the receptor is decreased the response to any agonist would be decreased.  This would also apply to the response to “physical stimuli”, as the authors classify cold and LPS. Therefore, I’m unsure how the data reported in this manuscript is novel or significant.

The authors attempt to suggest that other mechanisms may play a role, such as binding of LPS to the plasma membrane and association with lipid rats, which would increase its proximity to TRPA1.  Again, this does not matter if the expression of the channel is also decreased. 

The question that I, and most readers, would be interested in is if cholesterol depletion alters the single channel response in response to cold or LPS. This is probably best demonstrated using patch clamp.  However, this might also be accomplished by normalizing each cell's response to cold or LPS by its response to AITC, which would correct for the decrease in receptor expression.  If the normalized signal in response to cold or LPS still decreases, we can hypothesize that there is a direct effect of cholesterol depletion on the channel itself. 

Ans/ We thank this Reviewer for these comments. We would like to point out that our previous eLife paper reported that cholesterol depletion has a dual effect on TRPA1 responses, a reduction in the channel expression at the plasma membrane and a decrease in the sensitivity to the electrophilic agonist AITC (increased EC50). Thus, it is not straightforward to predict what the total effect of cholesterol depletion would be on cold- and LPS-induced activation, and other effects, in addition to a potential decrease in channel expression are not to be neglected.

Indeed, single-channel experiments would provide additional information, but we agree with the Reviewer in that this would imply a major experimental undertaking. Thus, we considered his/her suggestion to test whether cholesterol depletion induces differential effects on the responses to the different channel stimuli, which could be taken as proof that not only a decrease in channel expression at the plasma membrane is involved.

For the suggested normalization analysis to work, the amplitudes of the responses to cold or LPS should be correlated with the amplitudes of the responses to AITC in all experimental conditions. However, we have previously shown that there is a lack of correlation between the amplitudes of TRP channel-mediated responses to weak and strong stimuli (please, see Alpizar et al., Cell Calcium. 2013 Nov;54(5):362-74). In the present study we find back the same issue, because cold and LPS are relatively weak agonists of TRPA1 (see lines 90-93).

While trying to solve this problem, we noticed that we still needed to determine whether the reduction of channel expression induced by MCD we previously reported in transfected HEK293T cells is also seen in CHO-TRPA1 cells. To do this, we performed new functional experiments, to compare the concentration-dependence curves for AITC determined in control and after the MCD treatment, and to see whether the treatment reduces the maximal response to this compound. We tested AITC at concentrations up to 200 µM because at higher values we have seen an additional inhibitory effect (Everaerts et al., Current Biology, 2011). We found that the treatment with MCD increased the EC50 (as reported in our previous study in HEK293 cells), but to our surprise the maximal responses to AITC were not different between control and MCD-treated cells. This indicates that the expression of TRPA1 is not altered by the reduction of cholesterol in the CHO stable expression system, and that the most salient effect is that on the channel gating. This is now explained in the new Results section 2.3. Cholesterol depletion reduces sensitivity but not the maximal response of mTRPA1 to AITC. See lines 143-157 and Figure 5.

Another indication that the reduction in responses is not mediated by a decrease in channel expression at the plasma membrane is that the effects of MCD and SMase were weaker for LPS than for cold and AITC. See lines 133-134, 137-142 in the Results, and 152-157 in the Discussion.

These data introduce an extra element of novelty, and further justifies testing the effects of cholesterol reduction on the action of channel stimuli of distinct nature.

In all of the figures, the amplitude of the effects do not seem to match up with the reported change in intracellular calcium in the bar graphs. For example, in Figure 1, the amplitude of the calcium transients appears to all be above 1uM, yet the average reported change in calcium around 0.4uM.  The authors should show traces that better represent the reported averages. 

Ans/ Please, see in our new figures 1 to 4 that the amplitude values are spread over wide ranges. Therefore, we want to show traces of different amplitudes to give the reader an idea of how such traces look like. We belief that this is more important than showing several similar traces. In addition, given the recurrent discussion in this field, we do like to show traces with relatively high amplitudes to illustrate that TRPA1 responses to cold can be very robust.

The authors should list the exact number of cells for each experiment instead of the general “n>50” listing they have now.

Ans/ As requested, the cells numbers are now explicitly mentioned in the figure legends.

Round 2

Reviewer 3 Report

The authors have adequately addressed my earlier concerns.